# End-to-End Bubble Size Distribution Detection Technique in Dense Bubbly Flows Based on You Only Look Once Architecture

**DOI:** 10.3390/s23146582

**Published:** 2023-07-21

**Authors:** Mengchi Chen, Cheng Zhang, Wen Yang, Suyi Zhang, Wenjun Huang

**Affiliations:** 1College of Control Science and Engineering, Zhejiang University, Hangzhou 310027, China; 21632069@zju.edu.cn (M.C.); 22160075@zju.edu.cn (W.Y.); 2Luzhou Laojiao Co., Ltd., Luzhou 646000, China; zhangcheng@lzlj.com (C.Z.); zhangsy@lzlj.com (S.Z.)

**Keywords:** bubble size distribution, dense bubbly flows, end-to-end detector, objection detection, ellipse parameter fitting, *L*_2_ constraints

## Abstract

Accurate measurements of the bubble size distribution (BSD) are crucial for investigating gas–liquid mass transfer mechanisms and describing the characteristics of chemical production. However, measuring the BSD in high-density bubbly flows remains challenging due to limited image algorithms and high data densities. Therefore, an end-to-end BSD detection method in dense bubbly flows based on deep learning is proposed in this paper. The bubble detector locates the positions of dense bubbles utilizing objection detection networks and simultaneously performs ellipse parameter fitting to measure the size of the bubbles. Different You Only Look Once (YOLO) architectures are compared, and YOLOv7 is selected as the backbone network. The complete intersection over union calculation method is modified by the circumferential horizontal rectangle of bubbles, and the loss function is optimized by adding $L_{2}$ constraints of ellipse size parameters. The experimental results show that the proposed technique surpasses existing methods in terms of precision, recall, and mean square error, achieving values of 0.9871, 0.8725, and 3.8299, respectively. The proposed technique demonstrates high efficiency and accuracy when measuring BSDs in high-density bubbly flows and has the potential for practical applications.

## 1. Introduction

Bubbly flow is a common gas–liquid two-phase flow pattern which widely exists in water conservancy, petroleum, the chemical industry, the nuclear industry, and other fields. For example, the shape and size distributions of bubbles have a high correlation with the performance of mineral froth flotation [1]. The morphology of bubbles in Chinese spirits is closely related to alcohol concentration and quality [2]. Monitoring bubble parameters in hysteroscopic images is especially necessary to diagnose gas embolisms [3]. Detailed knowledge of bubble characteristics and dynamics is essential for process optimization. The interaction between the bubbles and the liquid determines the quantity of transferred mixing energy. Having an accurate measurement of the bubble size distribution (BSD) is very important for numerical flow modeling, the study of the gas–liquid mass transfer mechanism, and the design and control of gas–liquid multiphase reactors.

There are different BSD measurement methods for fluid mediums in the existing literature. They can be divided into intrusive and non-intrusive methods. Intrusive methods, such as capillary suction probes [4], conductivity probes [5], optical fiber probes [6], and wire-mesh sensors [7], are usually limited to a single or a few bubbles. Thus, the results may be biased by the applied measurement technique. Intrusive methods can be applied to a broader range of setups and with different liquids. Non-intrusive methods include phase doppler anemometry [8], interferometric particle imaging [9], and digital image analysis (DIA). Among non-invasive methods, direct imaging is the most commonly used since multiple objects can be observed simultaneously and the equipment is relatively cheap and flexible.

Traditional image processing techniques usually start with boundary extraction—either applying global or local thresholds—background subtraction, or edge detection by analyzing local intensity gradients. In the second stage, the boundary of the object, which can be individual bubbles or a cluster of bubbles, is segmented into curved segments of the bubbles. This can be achieved through seed-point extraction or breakpoint-based methods. Seed-points are defined as the centers of overlapping objects and are used to determine the number of bubbles in a cluster [10]. Different techniques are used, namely bounded erosion with elliptical distance transform [11], ultimate erosion [12], and fast radial symmetry transform [13]. The watershedding technique [14] and polygonal approximation method [15] are proposed to separate bubble objects into solitary bubbles and overlapping bubbles. Breakpoint methods segment the contours of clusters into contours of individual objects by detecting points where contours intersect. This can be done by analyzing the convexity of the contour [16] or by a combination of boundary curvature and intensity gradient analysis [17]. Several methods for reconstructing overlapping bubbles are proposed. An improved Hough transform algorithm can be used to reconstruct the contours of bubbles [18]. Ellipse parameters are estimated as bubble parameters using a stable direct least square fitting method [15].

A major problem with traditional imaging techniques is that the parameters of the different stages and even the workflow itself strongly depend on the experimental setup in which the images were taken. Experimental variables such as the distance between bubbles and the camera or the background light intensity significantly affect the applicability of these techniques. However, these variables are often limited by experimental conditions. Therefore, the potential for generalizing workflows is limited, requiring extensive manual adaptations and expertise in image processing.

Deep learning has become a prevailing machine vision technique due to the development of deep convolutional neural networks (CNN) in recent years. By comprehensively comparing the performance of various bubble detection methods when processing industrial image data, it is proved that the CNN-based method is superior to the circular Hough transform, concentric circular arrangements, and boosting-based detection methods [19]. Based on the deep learning algorithm, bubble center detection and synthetic bubble image generation are proposed to determine overlapping, blurred, and non-spherical bubble images [20]. A convolutional denoising auto-encoder architecture was introduced to recognize bubble patterns in images and identify their geometric parameters. The training dataset used synthetic images similar to real photos collected in experiments [21]. BubCNN employed a faster region-based CNN (RCNN) detector to localize bubbles and a shape regression CNN to predict bubble shape parameters [22]. Compared with state-of-the-art image processing procedures, BubCNN shows better generalization abilities. While the accuracy of BubCNN is high when the gas holdup is below 2%, its accuracy drops significantly as the gas holdup increases. Cerqueira and Paladino [23] developed a CNN-based method to identify bubbles and reconstruct bubble shapes in millimeter bubbly flows and obtained accurate results at gas holdups below 9%. A framework combining a deep edge-aware network and a marker-controlled watershed algorithm was proposed for extracting bubble parameters from hysteroscopy images. The proposed edge-aware network consists of an encoder–decoder architecture for bubble segmentation and a contour branch that is supervised by edge losses [3].

Researchers have also made great efforts in BSD detection in froth flotation. McGill bubble size analyzers (BSA) [24] are known to be the most acceptable techniques in both batch and industrial conditions. The Anglo Platinum Bubble Sizer was developed by the University of Cape Town in collaboration with Anglo Platinum as a modification of McGill BSA. Ma et al. [25] used a linear relationship between the bubble size and the area of the bright centroid to improve BSD characterization in the spherical regime in the presence of dense binding clusters. Bubble sampling is also critical to reduce the presence of clusters. Azgomi et al. [26] measured bubble size in a laboratory-scale flotation column which was designed with an expansion in the section. The results showed that the gas holdup decrease in the sampling point was effective in reducing the bubbles observed in the visual field. A new deflecting system that allowed a fraction of the sampled bubbles to be photographed was designed to reduce clusters of bubbles in the visual field [27]. An online BSD monitoring scheme was proposed by incorporating a multiscale-deblurring full convolutional network (FulConNet) (MsD) and a multistage jumping feature-fused FulConNet (MsJ), having the potential of online identification of the health state of the flotation process operations [28].

Although the current deep learning technology has improved the accuracy and robustness of bubble detection methods, it is worthwhile to mention that some problems remain. Firstly, the characteristics of bubbly flows with high gas holdup, such as low transparency, overlapping bubbles, and significant variations in bubble scale, impede the practical implementation of conventional bubble detection methods. Deep-learning-based methods in the literature are still limited to applications in gas–liquid flow with low gas holdups. Secondly, the current CNN models are typically designed as two-stage methods involving segmentation and reconstruction, which require a huge amount of computation and can impact the efficiency of the detection process.

This work proposes an end-to-end BSD detection method for dense bubbly flows. The proposed model optimizes the output structure and loss function, allowing it to effectively locate bubbles and determine size parameters through ellipse fitting simultaneously. The design of the loss function takes into account both localization loss and size fitting loss, thereby improving the accuracy of both bubble localization and size regression. Efficient objection detection model YOLO series models are researched and adopted as backbone models. This paper is structured as follows. In Section 2, we give a detailed description of the proposed method. In Section 3, we present the experimental results and analysis. Section 4 discusses the advantages and disadvantages of the proposed method. Finally, the conclusions are presented in Section 5.

## 2. Methodology

### 2.1. Overview

In contrast to previous tea-stage methods, an end-to-end BSD detection scheme based on deep learning is proposed in this paper, as shown in Figure 1. This scheme uses the objection detection framework to realize bubble location and size detection in the dense flow simultaneously. The whole process is divided into two stages of training and detection. During training, a bubble detector based on a deep CNN model is trained on public bubble datasets [29]. The model parameters are updated via backpropagation. During detection, the well-trained detector model takes bubble images as input, then performs the forward propagation, and finally outputs the bubble parameters.

In the field of computer vision, object detection includes two tasks, namely object location and object classification. Object classification is a classification problem to determine what category the target belongs to. Object location is essentially a regression problem that determines the coordinate position and size (usually the length and width of the bounding box) of the targets in the image. In this paper, the proposed method is only responsible for detecting a single category which is the “bubble” and does not distinguish the properties of bubbles (such as bubble shape, bubble color, etc.). Hence, the detector model ignores the classification requirement. BSD detection is essentially a regression prediction of the location and size parameters of the bubbles.

In many processes, the shape of rising gas bubbles can be approximated by an ellipsoid [22]. In a 2D projection, ellipsoids become ellipses, which are described mathematically by
(1)$$\frac{{\left[\left(x-x_{c}\right)\cos\left(\theta \right)+\left(y-y_{c}\right)\sin\left(\theta \right)\right]}^{2}}{a^{2}}+\frac{{\left[\left(x-x_{c}\right)\sin\left(\theta \right)+\left(y-y_{c}\right)\cos\left(\theta \right)\right]}^{2}}{b^{2}}=1$$
where $\left(x_{c},y_{c}\right)$ is the ellipse center coordinate, and a,b are the semi-major and semi-minor axes of the ellipse, respectively. θ is the anticlockwise rotation angle. It is worth noting that the location parameters refer to $\left(x_{c},y_{c}\right)$, and the size parameters refer to $\left(a,b,\theta \right)$. The detailed processing tricks are explained below.

### 2.2. BSD Detector Architecture

The proposed BSD detection technique is based on the You Only Look Once (YOLO) algorithm [30,31,32]. YOLO series models are state-of-the-art models for real-time object detection with a fast network architecture, effective feature integration methods, and robust loss functions. YOLO divides the image into grids and predicts bounding boxes and probabilities for each grid simultaneously.

The one-stage BSD detector architecture is shown in Figure 2, including a backbone, a neck, and multi-head modules. The backbone is the feature extraction network with stacked convolution, pooling, and other operation layers, such as ResNet-50 [33], Darknet53 [30], CSPDarknet53 [31], etc. CNN modules in the backbone finally downsample the input by 32.

Nick is composed of several bottom-up paths and several top-down paths to collect feature maps from different stages, such as SPP [34], FPN [35], etc. To enhance the detection probability of different bubble targets, upsampling is carried out twice. This operation contributes to obtaining more meaningful semantic features and finer-grained information from the earlier feature map.

Multi-head detection is utilized on three feature maps of varying scales, downsampled by factors of 8, 16, and 32, respectively. The backbone feature extractor is augmented with multiple convolutional layers. The last layer outputs a three dimensional tensor encoding ellipse parameters and confidence. Figure 3 illustrates that each detection output tensor is encoded as S×S×(B×(5+1)). This encoding is used because each output contains five ellipse parameters and one confidence parameter. *S* is the grid size of the feature map for each sub-detection path, and *B* is the anchor number for each path. It is worth noting that $t_{x},t_{y},t_{a},t_{b}$ are the offsets on the feature map instead of the final results, which is explained below.

### 2.3. Ellipse Location and Fitting

The initial approach of object detection typically employs the mean square error (MSE) for direct regression on the coordinates of the center point, as well as the height and width of the bounding box. Anchor-based methods estimate the corresponding offsets. Referring to the bounding box prediction method in YOLOv5 [36], the proposed model also utilizes the anchor-based strategy to detect bubbles. The difference lies in the design, where the detection network indirectly predicts four coordinates offsets $\left(t_{x},t_{y},t_{a},t_{b}\right)$ for each bubble, as shown in Figure 4. The final result is obtained using the following equations.
(2)$$x_{c}=\left(2\sigma \left(t_{x}\right)-0.5\right)+c_{x}$$
(3)$$y_{c}=\left(2\sigma \left(t_{y}\right)-0.5\right)+c_{y}$$
(4)$$a=p_{a}{\left(2\sigma \left(t_{a}\right)\right)}^{2}$$
(5)$$b=p_{b}{\left(2\sigma \left(t_{b}\right)\right)}^{2}$$
(6)$$bw=2\sqrt{a^{2}\cos^{2}\left(\theta \right)+b^{2}\sin^{2}\left(\theta \right)}$$
(7)$$bh=2\sqrt{b^{2}\cos^{2}\left(\theta \right)+a^{2}\sin^{2}\left(\theta \right)}$$
where $\left(c_{x},\;c_{y}\right)$ is the cell offset from the top left corner of the feature map. $p_{a},\;p_{b}$ are the prior semi-major and semi-minor axes of bubbles which are calculated by K-means [37]. In order to calculate intersection over union (IoU), the circumscribed horizontal rectangle of the rotated ellipse is used to replace the bounding box. bw and bh refer to the width and height of the circumscribed horizontal rectangle. All of the four offset parameters are predicted by a sigmoid function σ(). The paper employs the pixel as the unit of $x_{c},y_{c},a,b$ and radians as the unit of θ.

### 2.4. Loss Function Design

As described above, the backpropagation algorithm employs a loss function to constrain and guide the optimization of the model. We design a comprehensive loss function that includes location loss, regression loss, and confidence loss. Location loss $L_{loc}$ is used to guide optimization of parameters $t_{x}$ and $t_{y}$, regression loss $L_{reg}$ is used to fit ellipse parameters $t_{a}$, $t_{b}$, and θ, and confidence loss $L_{conf}$ is used to evaluate an objectness score for each bubble. The entire loss function Loss is defined as
(8)$$Loss=\lambda_{l}L_{loc}+\lambda_{r}L_{reg}+\lambda_{c}L_{conf}$$
(9)$$L_{loc}=1-CIoU$$
(10)$$L_{reg}=\sum_{i=0}^{S^{2}}\sum_{j=0}^{B}I_{ij}^{obj}\left[{\left(a_{ij}-{\hat{a}}_{ij}\right)}^{2}+{\left(b_{ij}-{\hat{b}}_{ij}\right)}^{2}+{\left(\theta_{ij}-{\hat{\theta }}_{ij}\right)}^{2}\right]$$
(11)$$L_{conf}=-\frac{1}{N}\sum_{i=0}^{S^{2}}\sum_{j=0}^{B}\left[o_{ij}\ln\left(conf_{ij}\right)+\left(1-o_{ij}\right)\ln\left(1-conf_{ij}\right)\right]$$
where $\lambda_{l},\lambda_{r},\lambda_{c}$ are constant factors. We use complete-IoU (CIoU) [38] to calculate location loss. Here, the bounding box is replaced by $\left(x_{c},y_{c},bw,bh\right)$. Regression loss is calculated by the mean square error loss. $I_{i}^{obj}$ denotes if an object appears in cell *i*, and $I_{ij}^{obj}$ denotes that the *j*th target in cell *i* is responsible for that detection. Confidence loss is calculated by binary cross-entropy loss. $a_{ij},b_{ij},\theta_{ij}$ are the predicted values, and ${\hat{a}}_{ij},{\hat{b}}_{ij},{\hat{\theta }}_{ij}$ are the true values. $o_{ij}\in \left[0,1\right]$ refers to the CIoU of the predicted target bounding box and real target bounding box, $conf_{ij}$ is the prediction confidence obtained by sigmoid function, and *N* is the total number of positive and negative samples.

## 3. Experiments

### 3.1. Dataset

All images in the dataset were synthetically generated using the public BubGAN [29] tool. BubGAN can be used for synthetic bubbly flow generation with customized bubbly flow boundary conditions. A total of one million synthetic bubbles are stored in the MillionBubble database with known bubble properties in terms of aspect ratio, rotation angle, circularity, and edge ratio.

In this work, 100,000 synthetic bubbly flow images were generated by assembling single bubbles in MillionBubble on an image background canvas with a size of 600×600 pixels. A total of 8000 images in the dataset are used as the training set, and the remaining 2000 images are used as the test set. The key parameters for generating images are bubble center coordinate $\left(x_{c},y_{c}\right)$, semi-major and semi-minor axes $\left(a,b\right)$, rotation angle θ, and void fraction. The distribution of the parameters is shown in Figure 5.

The center coordinates are subject to a random uniform distribution within the size of the canvas. The rotation angles are normally distributed. The semi-major axis and the semi-minor axis are mainly randomly distributed in $\left[5,40\right]$ pixels. The images have a resolution of around 25 pixels per mm. Bubbles are generally easy to detect in spherical regimes, while the detectability becomes more challenging when large bubbles coexist with small bubbles [39]. The rotation angle follows a Gaussian distribution, and its range is $\left[-\pi /2,\pi /2\right]$. The void fraction is defined by the volume of all bubbles divided by the total volume, which is randomly and uniformly distributed in $\left[0.01,0.1\right]$. The larger the void fraction is, the denser the bubbles become. Note that the void fraction in the test set is chosen from $\left\{0.01,0.02,0.03,0.04,0.05,0.06,0.07,0.08,0.09,0.1\right\}$, and 200 images are generated under each value.

### 3.2. Evaluation Metrics

In this work, the flowing several metrics are used to evaluate the trained detectors.

(1)Precision: The fraction of relevant instances among the retrieved instances
(12)$$\mathrm{Precision}=\frac{\mathrm{True}\;\mathrm{Positive}}{\mathrm{True}\;\mathrm{Positive}\;+\;\mathrm{False}\;\mathrm{Positive}}$$(2)Recall: The fraction of relevant instances that were retrieved
(13)$$\mathrm{Recall}=\frac{\mathrm{True}\;\mathrm{Positive}}{\mathrm{True}\;\mathrm{Positive}\;+\;\mathrm{False}\;\mathrm{Negtaive}}$$(3)AP50: the average precision (AP) value calculated at the threshold of 50% for detection evaluation. Specially, average precision is equivalent to mean average precision (mAP) due to a single-class detection task.(4)F1 score: A harmonic average of precision and recall
(14)$$F1=2\times \frac{\mathrm{Precision}\times \mathrm{Recall}}{\mathrm{Precision}\;+\;\mathrm{Recall}}$$(5)MSE: Average squared difference between the estimated values and the actual value.
(15)$$\begin{array}{c} \begin{array}{cc} \;\mathrm{MSE}\; & =\mathrm{MSE}\left(x_{c},y_{c},a,b,\theta \right)\\ \; & =\frac{1}{M}\sum_{i=1}^{M}\left[{\left(x_{ci}-{\hat{x}}_{ci}\right)}^{2}+{\left(y_{ci}-{\hat{y}}_{ci}\right)}^{2}+{\left(a_{i}-{\hat{a}}_{i}\right)}^{2}+{\left(b_{i}-{\hat{b}}_{i}\right)}^{2}+{\left(\theta_{i}-{\hat{\theta }}_{i}\right)}^{2}\right] \end{array} \end{array}$$
where *M* is the number of true positive samples, ${x_{c}}_{i},{y_{c}}_{i},a_{i},b_{i},\theta_{i}$ are the predicted values, and ${\hat{x}}_{ci},{\hat{y}}_{ci},{\hat{a}}_{i},{\hat{b}}_{i},{\hat{\theta }}_{i}$ are the true values.(6)Frames Per Second (FPS): the frequency (rate) at which consecutive images (frames) are inferred(7)FLOPs: Floating point operations to measure the complexity of the CNN model.

The metrics of precision, recall, AP50, and F1 score are used to evaluate the model’s ability in bubble detection, while MES is used to evaluate the model’s ability to fit elliptical parameters.

### 3.3. Analysis of Backbone Networks

In this study, we initially assess the efficacy of our approach by testing different backbone network structures to identify the optimal model architecture. We analyzed and compared 11 mainstream YOLO networks, including YOLOv3 [30], YOLOv3-spp [30], YOLOv3-tiny [30], YOLOv4-L [31], YOLOv4-M [31], YOLOv4-S [31], YOLOv5-L [36], YOLOv5-M [36], YOLOv5-S [36], YOLOv7 [32], and YOLOv7-tiny [32]. It should be noted that during the experiment, we only used the network architectures of different YOLO versions as backbone networks. However, some tricks, such as CutMix and Mosaic data augmentation in YOLOv4, were not exploited, and the method used to predict position parameters is different from that used in the original paper.

We utilized a server equipped with an NVIDIA TITAN Xp GPU to train and test the model. During the model training process, the following parameter settings were utilized: input size = 640×640×3; the optimization algorithm was set to Adam; batch size = 32; epochs = 100; and initial learning rate = $1\times 10^{-3}$. Additionally, we introduced weight decay with a factor of $1\times 10^{-4}$ to tackle overfitting. The three constant coefficients in the loss function were set to $\lambda_{l}=0.5$, $\lambda_{r}=0.5$, and $\lambda_{c}=0.1$. The IoU threshold for training was set to 0.2.

The model performs multi-head detection on three different feature maps (three branches) of different sizes, which are 80×80, 40×40, and 20×20. Each detection branch has three anchors, resulting in a total of $B=9\left(3\times 3\right)$ anchors. Therefore, the K-means algorithm is used to cluster the parameters in the training set, with a total of nine clusters. The clustering results are (13, 10), (26, 20), (33, 28), (40, 36), (45, 25), (48, 43), (54, 34), (56, 51), and (72, 64). Due to the consistent background generated in the dataset, random flipping is used as the only data augmentation method during training.

Since BubCNN [22] is considered to be the state-of-the-art method for bubble detection, we conducted a comparative analysis between our method and BubCNN. BubCNN was trained for 50 epochs with an initial learning rate of $1\times 10^{-4}$. The learning rate was halved every 10 epochs. During training, 1000 region proposals were evaluated, and the overlap thresholds IoU${}_{\max}$ and IoU${}_{\min}$ were set to 0.7 and 0.3, respectively.

Based on the evaluation metrics defined in the previous section, we compared the performance of 11 different backbone network models as shown in Table 1. It can be seen that the precision of 11 different backbone network models is above 0.97, among which the precision of YOLOv5-L backbone network model is the highest, reaching 0.9879. The model with the lowest precision is the one in which the backbone network is YOLOv4-S, and the precision is 0.9750. The precision of these two models differs only by 0.0129. The average precision of the 11 different backbone models is 0.9817, and the variance is $2.12\times 10^{-5}$. Due to the absence of noise interference in the dataset images used in this paper, all models exhibit high precision with minimal differences. The precision of BubCNN is 0.8130. Compared with BubCNN, the precision of YOLOv5-L model is increased by 0.1749, and the precision of YOLOv4-S model is increased by 0.1620.

The difficulty studied in this paper is the detection of bubbles under dense conditions, so recall is a more important metric to focus on. the recall of the model with YOLOv7 as the backbone network is the highest among all models, reaching 0.8725. The recall of the model with YOLOv3-tiny as the backbone model is the lowest, with a value of 0.8310. The recall of the YOLOv7 model differs from that of the YOLOv3-tiny model by only 0.0415. The average recall rate of the 11 different backbone models is 0.8598, and the variance is $1\times 10^{-4}$. Although the precision differences of models with different backbone networks are relatively small, there are still significant differences in the recall rates. The recall rate of BubCNN is only 0.5100, which is increased by 0.3625 when compared to YOLOv7, and increased by 0.3210 when compared to YOLOv3-tiny. This indicates that the method proposed in this paper has good detection performance for bubbles under dense conditions.

The advantage of the AP metric is that it considers a balance between precision and recall at different confidence threshold levels and can be used to compare the performance of different models. The F1 score is a statistical measure used to evaluate the precision of binary classification models, as it considers both precision and recall. Based on the analysis of the AP50 and the F1 score, the best-performing model among all the tested models is the one that uses the YOLOv7 backbone network, with AP50 = 0.9161 and F1 score = 0.9263. YOLOv3 uses Darknet53 as the backbone network. YOLOv4 adds the CSPDarknet53 structure and the Mish activation function on the basis of YOLOv3. YOLOv5 stacks the ConvBNSiLU module, C3 module, and SPPF module to form the backbone, and the YOLOv7 backbone network is composed of the CBS module, ELAN module, and MPConv module. The experimental results indicate that the network structure design of YOLOv7 is more suitable for the detection of dense bubbles. The AP50 result of the YOLOv7 model surpasses BubCNN (0.6440) by 42%, demonstrating the superior performance of the method proposed in this paper for bubble localization.

The MSE metric is an overall evaluation of the model’s fitting performance for two positional parameters $x_{c},y_{c}$ and three size parameters a,b,θ. From the data in Table 1, the model that has the best fitting performance is the one that uses the YOLOv4-L network, with an MSE error of 3.7273 (in pixels). However, the YOLOv4-L model has average results in terms of precision, recall, AP50, and other metrics. The model with the largest error is the one based on the YOLOv3-tiny structure, with an MSE of 7.2400, due to the shallowest layer of the YOLOv3-tiny model. The average MSE of 11 different backbone models is 4.6765, with a variance of 0.9621. In terms of precision, recall rate, AP50, F1 score, and MSE, the distribution of model performance on the MSE metric is the largest, because the fitting of position and size is evaluated at the pixel level, while other metrics consider predicted results as true positive samples when the IoU between the predicted result and the label is larger than 0.5. The MSE of the YOLOv7 model is 3.8299, which is only 0.1026 higher than the best result, and it is 9.3701 higher than the MSE of BubCNN (13.2). Therefore, the performance of the model based on the YOLOv7 backbone network is excellent as well in terms of parameter fitting.

Taking all evaluation metrics into account, the results in Table 1 demonstrate that the proposed method has good detection performance and parameter fitting precision in dense bubbly flows. Under the premise of considering detection efficiency, it is believed that the model based on the YOLOv7 backbone network has the best overall performance, which is why the analysis of the loss function, IoU, and void fraction in the following sections are based on the YOLOv7 network.

### 3.4. Analysis of Loss Function and IoU

In object detection, the YOLO algorithm predicts the bounding box for targets through regression. Traditional object detection algorithms use the MSE to predict the coordinates of the center point as well as the length and width of the bounding box, or they predict the upper left and lower right points. With the evolution and iteration of the YOLO algorithm, researchers have used anchor-based methods to estimate the corresponding offset for the position. They use IoU loss to predict the offset of the bounding box relative to the anchor. As the parameters predicted in this study are not bounding boxes, it is necessary to redesign an optimal loss function.

The method proposed in this paper assumes that the bubble is an ellipse rotating counterclockwise around the *x*-axis. The loss function is composed of three parts: location loss $L_{loc}$, regression loss $L_{red}$, and confidence loss $L_{conf}$. The location loss is calculated in the form of CIoU loss.

The effects of loss coefficient $\lambda_{l}$, $\lambda_{r}$, and $\lambda_{c}$ for training were investigated. These three parameters were chosen based on empirical methods. Here, we conducted tests using various combinations selected from three values: 0.1, 0.5, and 1.0. The results are given in Table 2. In general, the selection of various combinations has a minor impact on both the location performance and the fitting accuracy of the model. When the values of coefficients $\lambda_{l}$ and $\lambda_{r}$ are small, it results in a slight increase in the MSE. This occurs because reducing these two coefficients weakens the constraints on the location and size parameters during the training process. Comparatively, we prefer to improve the location performance of the bubbles. We ultimately selected a parameter combination with the highest AP50 value, which corresponds to $\lambda_{l}=0.5$, $\lambda_{r}=0.5$, and $\lambda_{c}=0.1$.

When calculating IoU, the circumferential horizontal rectangle of the ellipse is taken as the target position. The center coordinates $\left(x_{c},y_{c}\right)$, width bw, and height bh of the circumferential horizontal rectangle are calculated by Equations (6) and (7), respectively. Theoretically, it is possible to train the model using only IoU loss through backpropagation to fit the five parameters $t_{x},t_{y},t_{a},t_{b},\theta$. However, in the experimental process, we found that only fitting the size parameters of the ellipse does not result in optimal performance. Therefore, we experimented with four different methods, including IoU [40], GIoU [41], DIoU [38], and CIoU [38]; the different regression losses are shown in Equations (16)–(18), respectively. The difference between the three different modes lies in whether to include the MSE of the size parameters $\left(a,b,\theta \right)$ in the loss function.
(16)$$Mode_{1}:L_{reg}=\sum_{i=0}^{S^{2}}\sum_{j=0}^{B}I_{ij}^{obj}\left[{\left(a_{ij}-{\hat{a}}_{ij}\right)}^{2}+{\left(b_{ij}-{\hat{b}}_{ij}\right)}^{2}+{\left(\theta_{ij}-{\hat{\theta }}_{ij}\right)}^{2}\right]$$
(17)$$Mode_{2}:L_{reg}=\sum_{i=0}^{S^{2}}\sum_{j=0}^{B}I_{ij}^{obj}\left[{\left(\theta_{ij}-{\hat{\theta }}_{ij}\right)}^{2}\right]$$
(18)$$Mode_{3}:L_{reg}=0$$

Table 3 presents the experimental results using different IoU and loss functions. The left-hand side represents bubble detection performance, while the right-hand side reflects bubble parameter fitting performance. When using $Mode_{1}$ to calculate the loss function, the four methods of IoU calculation have little impact on precision. The highest model precision is 0.9871 when using CIoU to calculate the loss function and lowest at 0.9859 when using DIoU. The small difference in precision indicates that almost all detected bubbles are true positives. However, the different IoU calculation methods have a certain impact on the model’s recall. Among them, the model using CIoU has the highest recall at 0.8725 and the best AP50 and F19 scores, reaching 0.9161 and 0.9263, respectively. The DIoU model follows, with a recall $2.7\times 10^{-4}$ lower than that of the CIoU model, and AP50 and F1 scores decreasing by $2.5\times 10^{-4}$ and $2.0\times 10^{-4}$, respectively. The IoU and GIoU methods have little difference in recall, AP50, and F1 score metrics. When all models use CIoU methods, more parameters considered for regression loss ($Mode_{1}$) resulted in better results in terms of precision, recall, AP50, F1 score, and other aspects. Overall, different loss functions have a relatively small difference in bubble detection performance, with slight fluctuations in numerical results but have a significant impact on the fitting of size parameters.

Analysis of the data in the right half of Table 3 shows that the design of regression loss is crucial for the accuracy of parameter fitting. Traditional object detection algorithms only use IoU to predict the bounding box parameters of objects. However, when using IoU loss only to fit the location and size parameters of bubbles in this paper, the fitting performance of the model is the worst, especially for size parameter fitting with significant deviations. Although the CIoU + $Mode_{3}$ method had the largest fitting MSE of location parameters, the difference was only under two, so the performance is considered to be at the same level as the other methods. However, the MSE of parameters a,b are 14.4 and 10.3, higher than the best model (CIoU + $Mode_{1}$) by 5.2 and 4.1, respectively. $Mode_{2}$ and $Mode_{3}$ did not calculate the $L_{2}$ loss of a,b , so the MSE of these two parameters were larger than those of other models. The CIoU + $Mode_{3}$ model had an MSE of 0.288 for the parameter θ, while the best model (CIoU + $Mode_{2}$) had a significantly lower MSE of only 0.0364, which was 7.9 times smaller. The MSEs of other models for the parameter were around 0.04 because they calculated the $L_{2}$ loss of θ. Therefore, calculating the $L_{2}$ loss of $\left(a,b,\theta \right)$ in regression loss can significantly improve the model’s parameter detection accuracy.

Through the experimental analysis in this section, we found that the optimal solution for the loss function is to use CIoU to calculate the location loss and $Mode_{1}$ to calculate the regression loss. Moreover, compared to BubCNN’s performance, the CIoU + $Mode_{1}$ method has shown significant improvements in all performance evaluation metrics, particularly in recall and MSE. In terms of bubble localization, precision increased by 0.1741, recall increased by 0.3625, AP50 increased by 0.2721, and F1 score increased by 0.2995. In terms of parameter fitting, the MSE of parameter $x_{c}$ and parameter $y_{c}$ decreased by 13.8 and 17.4, respectively. The MSE of parameter *a* and parameter *b* decreased by 5.7 and 9.7, respectively. MSE of parameter θ decreased by 0.4435, and the average MSE of all five parameters decreased by 9.4.

The design of the IoU loss function in object detection has the potential to accurately predict the bounding box of the target, which comprises four parameters. However, our research focuses on bubble objects that differ from traditional bounding boxes and necessitate the prediction of five parameters. The prediction of parameters $x_{c}$ and $y_{c}$ is equivalent to estimating the center coordinates of the bounding box. Therefore, the IoU loss can be employed. However, since the prediction of parameters *a*, *b*, and θ differs from the detection of bounding box width and height, the $L_{2}$ loss is employed to enforce the constraint. The optimized loss function offers the following advantages. Firstly, it fulfills the model’s requirement to simultaneously output both localization and size parameters. Secondly, it substantially enhances the accuracy of predicting the bubble’s size parameters, particularly parameter θ. Thirdly, it results in a marginal improvement in the localization performance of the bubbles.

### 3.5. Bubble Size Distribution Estimation Evaluation

This section evaluates the estimated results of the BSD from two perspectives: the number of bubbles and the size of the bubbles. Figure 6 illustrates the number of bubbles estimated from the test set of 2000 images. The blue curve represents the actual number of bubbles in each sample image, while the yellow curve represents the estimated number of bubbles. For better visualization, minor adjustments were made during the plotting process, including sorting the actual bubble quantities in ascending order. From the figure, it can be observed that when the number of bubbles in each image is less than approximately 60, the model can detect almost all of them. However, as the number of bubbles increases, the deviation between the predicted and actual values gradually becomes larger. The increase in bubble quantity implies a higher void fraction, resulting in more challenging bubble detection, which aligns with our intuitive understanding. The following sections will provide a detailed numerical analysis of precision and recall.

The pixel lengths of the semi-major axis *a* in the test set were divided into four intervals: 2<a≤10, 10<a≤20, 20<a≤30, and 30<a≤40. In physical space, every 25 pixels correspond to a length of 1 mm. The estimated results of BSD are shown in Figure 7, where the blue curve represents the actual values and the yellow curve represents the estimated values. Similarly, for the purpose of analysis, we sorted the bubbles in ascending order according to the actual value of *a*. Therefore, the actual value curves in Figure 7a,d,f,g exhibit a monotonically increasing trend. The analysis of MSE data in Table 4 reveals a clear trend: as the bubble size increases, the fitting errors for parameters *a* and *b* also increase. Notably, the MSE of parameter *b* is relatively smaller in comparison to parameter *a*. When 2<a≤10, the MSE values for *a* and *b* are only 0.5 and 0.3, respectively. However, when 30<a≤40, their MSE values increase to 9.2 and 6.5. This is due to the presence of densely distributed bubbles, where larger bubbles are more likely to overlap and occlude other bubbles. The coexistence of large and small bubbles introduces noise and impacts the analysis of the image. Consequently, larger bubble sizes result in larger errors. However, as the size of the bubbles increase, the MSE of parameter θ gradually decreases from 0.0796 to 0.0355. This is because when the bubble size is too small, the rotational effect is not visually significant, making detection difficult. As the bubble size increases, although interference from other bubbles may be present, this noise has a smaller impact on rotation.

### 3.6. Analysis under Different Void Fractions

Based on the previous experiment, we have explored the best comprehensive performance solution, which uses YOLOv7 as the backbone network and calculates the localization loss and regression loss separately using the CIoU method and Mode1 method. This solution has shown excellent performance in terms of bubble precision, recall, fitting accuracy, and efficiency. In this section, we will explore the detection results of this model under different void fraction conditions.

Figure 8 illustrates the detection results of the best-performing model under different void fractions. The void fraction represents the proportion of the area occupied by bubbles in two-phase flow. The larger the void fraction, the larger the area occupied by bubbles, and the denser the distribution. The void fraction ranges from 0.01 to 0.10 with a step size of 0.01. The images in Figure 8 represent typical pictures at each void fraction. The green dots in each subgraph represent the fitted elliptical bubble center, and the red edge represents the boundary of the ellipse. As evident from the figure, the proposed model accurately detects bubbles and fits elliptical parameters for both sparse and dense bubble distributions.

Table 5 shows the detection performance metrics of the model at different void fractions, with 200 test images at each void fraction. The results of precision, recall, AP50, and F1 score all decrease as the void fraction increases, with recall decreasing the most, from 0.9794 to 0.8120. Precision decreases the least, from 0.9988 to 0.9826, decreasing by 0.0162 in total. The greater decrease in recall indicates an increased rate of false positive detections as the bubbles become denser. As observed from Figure 8, the primary reason for this is that some bubbles are significantly occluded and cannot be detected when the bubbles become denser. The minor decrease in precision is caused by the detection of redundant objects in areas with heavily stacked clusters of bubbles. The multiple edges of tightly packed bubbles in clusters can visually resemble new bubbles, as shown in the top-left region of the bubble cluster in Figure 8h. The slightly decreasing trend of AP50 and F1 score by 0.0808 and 0.0998, respectively, suggests that the model still achieves excellent detection performance in dense bubble distributions.

From the overall perspective, the MSE increases as the void fraction increases. The increases in the MSE of $x_{c}$, $y_{c}$, *a*, *b* and θ are 2.0, 1.4, 3.3, 2.2, and 0.0304, respectively. Considered within the value ranges of the five parameters (as shown in Figure 9), the changes in the MSE increase relatively slowly. This suggests that the model’s fitting ability is robust and resilient. From Figure 8, it is observed that the reason for the increase in the MSE is due to the possibility of fragmented and overlapped bubbles with dense bubble distributions. The fitting results of these bubbles by the model may exhibit some bias, as seen in the top-right edge of the bubble in Figure 8d and the bottom-right edge of the bubble in Figure 8g.

Figure 9 presents the box plot of MSE errors for the parameter fitting. The figure indicates that the increase in void fraction has a minor impact on the median of the MSE errors for all five parameters, suggesting that the central distribution of fitting errors remains relatively unchanged and positively skewed. However, for parameters $x_{c}$, $y_{c}$, and θ, the increase in void fraction results in an increase in the length of the box, and the upper margin shows exponential growth, enhancing the positively skewed distribution. The upper edge of parameters *a* and *b* slightly increases with the increase in void fraction, and its error distribution is more stable in comparison to the other three parameters. The appearance of a higher upper edge indicates the occurrence of missed detections (false positives) in rare samples. Figure 9 further suggests that the fitting results for most bubbles are minimally affected by the increase in void fraction, but the main impact is on the occurrence of anomalous detection results.

Overall, the model proposed in this paper exhibits high accuracy and robustness in detecting dense bubbles and fitting parameters.

## 4. Discussions

The traditional image-processing-based bubble detection method can achieve good detection performance under specific experimental conditions. However, changes in the usage scenario often lead to a decrease in algorithm performance or even make it unsuitable for use. BubCNN was the first to apply the deep-learning-based object detection network Faster-RCNN to the field of bubble detection and combine it with a shape regression CNN network for bubble shape parameter detection to create a two-stage bubble detector. Following this path of object detection, this paper designs an end-to-end bubble detection network based on the YOLO network to achieve more efficient bubble location and parameter detection. Through experimental comparative analysis, our proposed method has the following advantages: Firstly, for densely distributed bubbles, the proposed model greatly improves the precision and recall over existing methods. The model has better overall bubble positioning performance and has smaller fitting errors for parameters. Secondly, the improved bubble detection network proposed, which is based on the YOLO method, is a one-stage network with higher detection efficiency and real-time performance. The fastest model achieves an FPS of 255. Lastly, the open-source BubGAN dataset was used for training the data. The diverse shapes of bubbles in the dataset enhance the model’s generalization ability and extend its applicability to a wider range.

Nevertheless, the proposed method also has some limitations. Firstly, in highly dense bubble distributions, the proposed method may produce false positive missed detections and erroneous parameter detection results for bubbles that are incomplete or fragmented at the edges of the image. In addition, although the assumption that the bubble contour is usually elliptical is valid, it becomes inaccurate when dealing with higher Eotvos and Reynolds numbers. Therefore, deep-learning-based instance segmentation methods can be explored as potential solutions to address these two issues. Bubbles are segmented with pixel-wise precision, and morphological parameters of each small bubble can be extracted as needed. Finally, it should be noted that this research is based on 2D bubble images, while 3D bubbles may lose some size features after being sampled by 2D cameras. The stereological correction was not taken into consideration to determine bubble size, which may result in certain errors.

## 5. Conclusions

To address the problem of difficult detection of bubbles in dense bubbly flow, this paper proposes a novel BSD detection method based on deep learning. The main contributions of this paper are as follows:(1)An end-to-end BSD detection method based on YOLO is proposed, using a multi-head detection method to detect different-sized bubbles from three scales, and detecting dense objects by using a dense detection approach. The model also adds an ellipse fitting output for the morphological parameters of the bubbles, achieving a synchronous output of their position and position parameters.(2)The loss function of dense bubble detection is optimized using CIoU of bubble objects and $L_{2}$ constraints of elliptical parameters to improve the accuracy of model parameter fitting. The precision, recall, and AP50 of the model are 0.9871, 0.8725, and 0.9161 respectively, and the MSE of the parameters is 3.8299.

Furthermore, pixel-level detection of bubbles will be researched in further work by using deep-learning-based instance segmentation techniques.

## Figures and Tables

**Figure 1 sensors-23-06582-f001:**
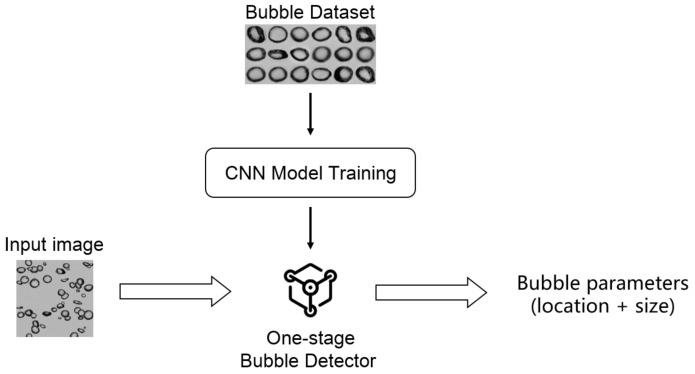
End-to-end BSD detection workflow.

**Figure 2 sensors-23-06582-f002:**
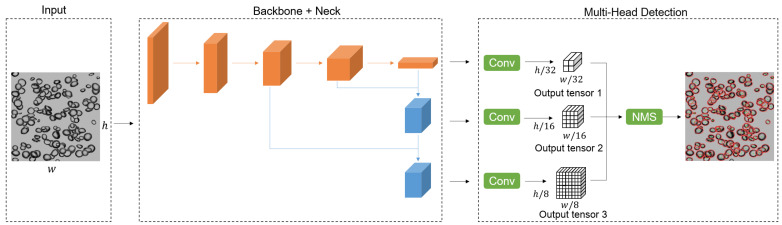
The proposed one-stage BSD detector architecture based on YOLO. The orange and blue blocks represent the backbone module and the neck module, respectively.

**Figure 3 sensors-23-06582-f003:**
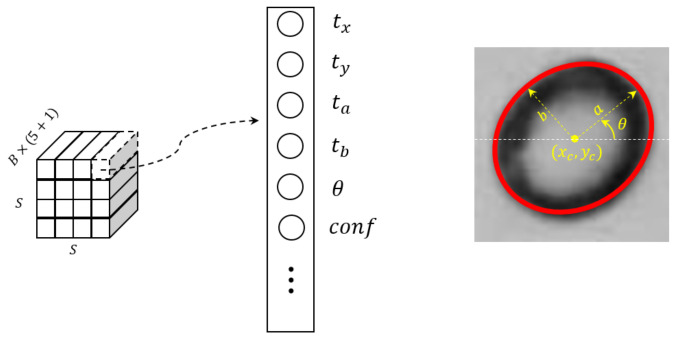
Detector output illustration.

**Figure 4 sensors-23-06582-f004:**
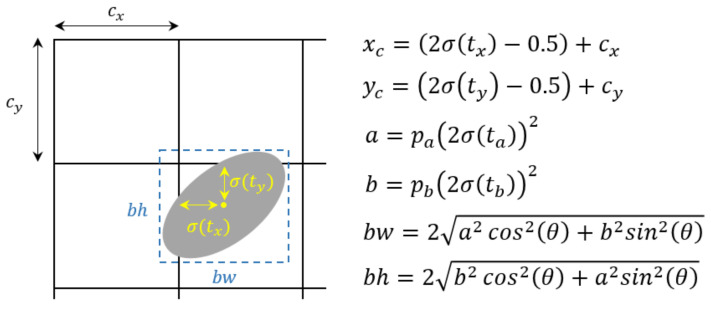
Ellipse fitting with dimension priors.

**Figure 5 sensors-23-06582-f005:**
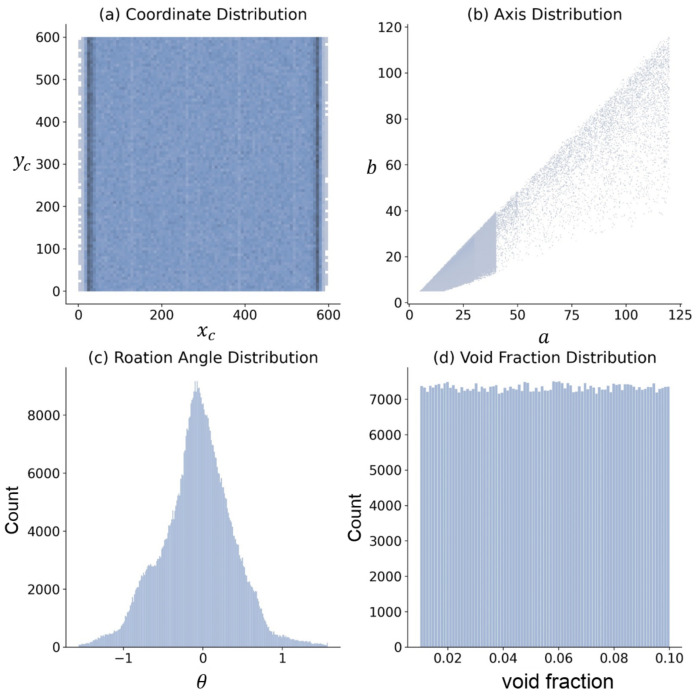
The distribution of dataset parameters.

**Figure 6 sensors-23-06582-f006:**
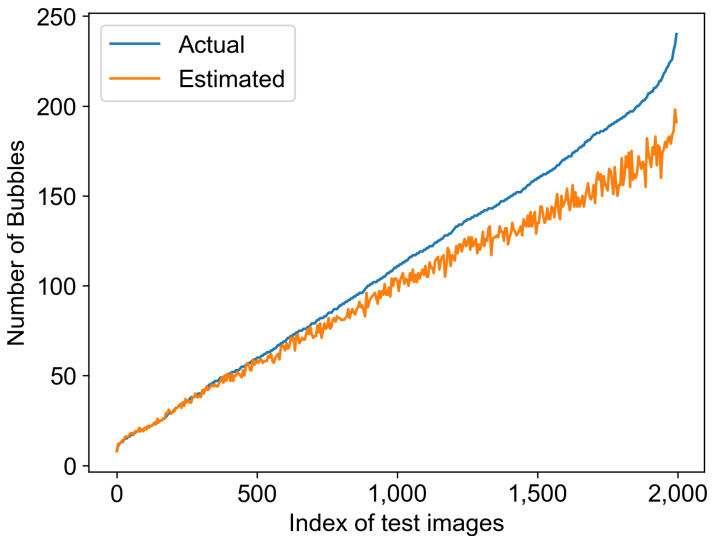
Comparison of the estimated and actual number of bubbles.

**Figure 7 sensors-23-06582-f007:**
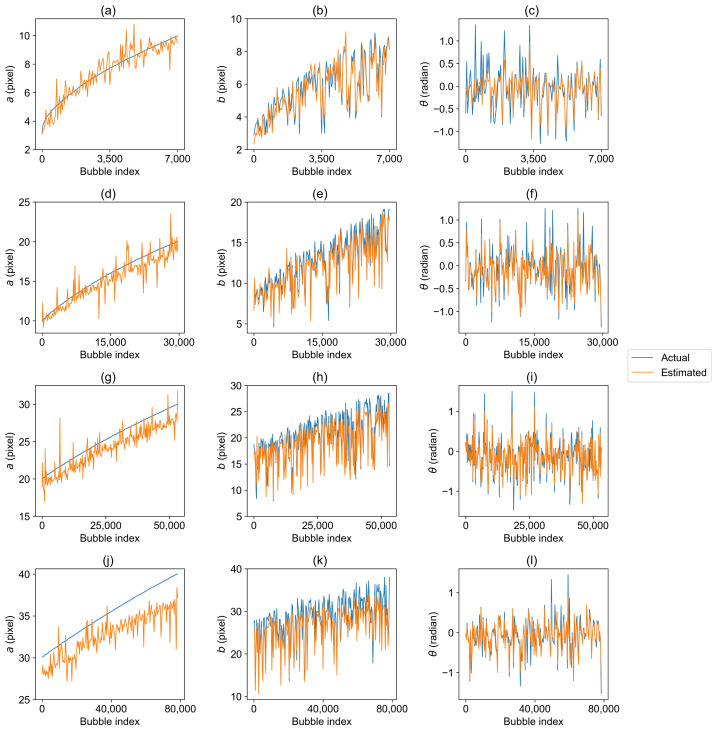
Comparison of the estimated and actual bubble size. The sub-figures in the first, second, and third columns display the comparison results between the estimated and actual values of *a*, *b*, and θ under different ranges of the semi-major axis (*a*). (**a**–**c**): 2<a≤10; (**d**–**f**): 10<a≤20; (**g**–**i**): 20<a≤30; (**j**–**l**): 30<a≤40.

**Figure 8 sensors-23-06582-f008:**
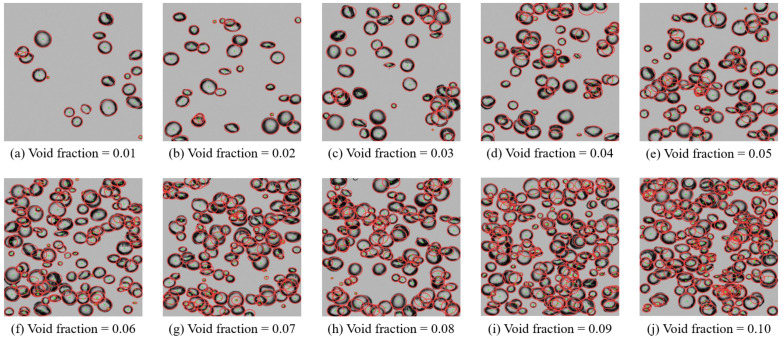
Detection results under different void fractions.

**Figure 9 sensors-23-06582-f009:**
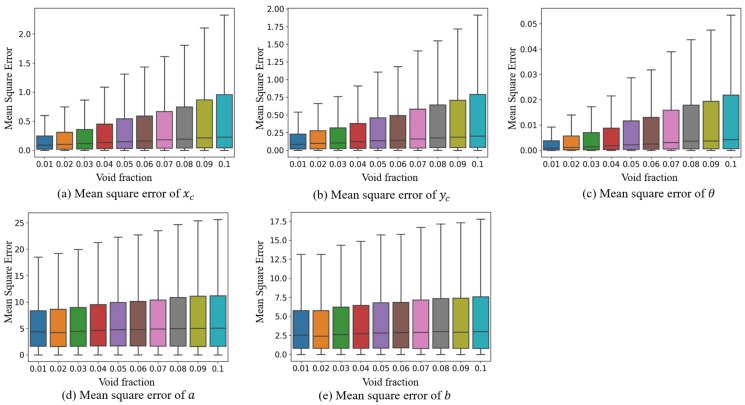
Mean square error results of different void fractions.

**Table 1 sensors-23-06582-t001:** Detector performance of different backbone networks.

Backbone	Precision	Recall	AP50	F1 Score	MSE	FLOPs	Parameters	FPS
YOLOv7	0.9871	0.8725	0.9161	0.9263	3.8299	103.2G	36,497,954	55
YOLOv7-tiny	0.9843	0.8614	0.9152	0.9187	4.6796	13.0G	6,015,714	144
YOLOv5-L	0.9879	0.864	0.915	0.9218	3.74	107.7G	46,124,433	51
YOLOv5-M	0.9826	0.863	0.9131	0.9189	4.6239	47.9G	20,865,057	76
YOLOv5-S	0.9831	0.854	0.9137	0.914	4.5461	15.8G	7,020,913	135
YOLOv4-L	0.984	0.862	0.9067	0.919	3.7273	119.1G	52,496,689	46
YOLOv4-M	0.986	0.865	0.9063	0.9215	4.1	50.3G	24,357,849	69
YOLOv4-S	0.975	0.861	0.903	0.9145	4.8645	20.6G	9,118,721	117
YOLOv3	0.976	0.8571	0.9116	0.9127	4.39	154.6G	61,513,585	41
YOLOv3-spp	0.977	0.867	0.9132	0.9187	5.7	155.4G	62,562,673	40
YOLOv3-tiny	0.976	0.831	0.903	0.8977	7.24	12.9G	8,673,622	250
BubCNN	0.813	0.51	0.644	0.6268	13.2	-	-	2

**Table 2 sensors-23-06582-t002:** Detector performance with different coefficients for the loss function.

λ1	$\lambda_{r}$	$\lambda_{c}$	Precision	Recall	AP50	F1 Score	MSE
0.5	0.5	0.5	0.9848	0.8738	0.9063	0.9260	3.9323
1.0	0.5	0.5	0.9853	0.8738	0.9067	0.9262	3.7166
0.5	1.0	0.5	0.9870	0.8713	0.9062	0.9255	3.8856
0.5	0.5	1.0	0.9842	0.8723	0.9058	0.9249	3.9255
0.1	0.5	0.5	0.9832	0.8738	0.9066	0.9252	4.0088
0.5	0.1	0.5	0.9859	0.8718	0.9135	0.9254	4.2846
0.5	0.5	0.1	0.9871	0.8725	0.9161	0.9263	3.8299

**Table 3 sensors-23-06582-t003:** Analysis of loss function and IOU.

Loss Function	Precision	Recall	AP50	F1 Score	MSE					
					$x_{c}$	$y_{c}$	*a*	*b*	θ	Total
IoU + $Mode_{1}$	0.9861	0.8654	0.9087	0.9218	2.4	1.8	11.3	7.4	0.0443	4.5852
GIoU + $Mode_{1}$	0.9863	0.8697	0.9093	0.9243	2.9	2.1	12.9	9.8	0.0449	5.5473
DIoU + $Mode_{1}$	0.9859	0.8699	0.9136	0.9243	2.3	1.7	10.1	7.2	0.0407	4.2846
CIoU + $Mode_{1}$	0.9871	0.8725	0.9161	0.9263	2.1	1.6	9.2	6.2	0.0385	3.8299
CIoU + $Mode_{2}$	0.9834	0.867	0.9103	0.9215	2.5	1.9	11.9	11.7	0.0364	5.6338
CIoU + $Mode_{3}$	0.9792	0.861	0.9116	0.9163	3.3	2.5	14.4	10.3	0.2888	6.1692
BubCNN	0.813	0.51	0.644	0.6268	15.9	19	14.9	15.9	0.4820	13.2364

**Table 4 sensors-23-06582-t004:** MSE of bubbles with different semi-major axis sizes.

Semi-Major Axis Range	2<a≤10	10<a≤20	20<a≤30	30<a≤40
MSE of *a*	0.5	1.5	3.7	9.2
MSE of *b*	0.3	1.0	2.6	6.5
MSE of θ	0.0796	0.0415	0.0348	0.0355

**Table 5 sensors-23-06582-t005:** Test results under different void fractions.

Void Fraction	Precision	Recall	AP50	F1 Score	MSE					
					$x_{c}$	$y_{c}$	*a*	*b*	θ	Total
0.01	0.9988	0.9794	0.9856	0.989	0.4	0.4	6.3	4.4	0.0165	2.2949
0.02	0.9956	0.9602	0.9765	0.9776	1	0.8	7	4.7	0.0234	2.7038
0.03	0.9962	0.9456	0.9704	0.9703	1.4	1	8	5.5	0.0253	3.1874
0.04	0.9925	0.9343	0.9637	0.9625	1.8	1.4	9	5.9	0.0288	3.6188
0.05	0.991	0.9116	0.9531	0.9496	2.2	1.6	9.7	6.4	0.0351	4.0073
0.06	0.9906	0.8905	0.9441	0.9379	2.4	1.7	9.5	6.3	0.0371	4.0003
0.07	0.9853	0.8747	0.936	0.9267	2.2	1.7	9.2	6.3	0.041	3.8744
0.08	0.9848	0.8571	0.9279	0.9165	2.2	1.6	9.4	6.4	0.0421	3.9475
0.09	0.9812	0.8354	0.9162	0.9024	2.4	1.8	9.5	6.4	0.0442	4.0237
0.1	0.9826	0.812	0.9048	0.8892	2.4	1.8	9.6	6.6	0.0469	4.1023

## Data Availability

Not applicable.

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
