# Peer review of "End-to-End Bubble Size Distribution Detection Technique in Dense Bubbly Flows Based on You Only Look Once Architecture"

_sensors, 2023, doi:10.3390/s23146582_

Round 1

Reviewer 1 Report

This is not the well written paper.

The paper has several punctuation mistakes. This paper is unstructured format as follows.

In line 92 they are incomplete sentences  In Section ??, the paper is not thorughly checked

references are incomplete 

This is not the well written paper.

The paper has several punctuation mistakes. This paper is unstructured format as follows.

In line 92 they are incomplete sentences  In Section ??, the paper is not thorughly checked

references are incomplete 

Reviewer 2 Report

This manuscript is interesting. I believe it is appropriate for publication  after some major revisions. The first main action which needs to be taken is to explain the processing procedure much more clearly. The second main action is to give more insight into how variable the image analysis results are. My full list of comments are below.

1. The introduction section must be improved. In particular, the problem of bubble detection must be presented, as well as a discussion of the state of the art. 

2. The novelty of the paper seems to be the loss function, but no justification/benefit of this setup is given.

3. I think one experiment is not enough. It is better to add another public dataset.

4. The parameters used are not justified or selected according to a systematic methodology. Any method used for comparison should also be described, and the parameters used should be reported.

A few of spelling and grammatical errors

Reviewer 3 Report

The manuscript presents a substantial progress in objection detection as with the developed YOLO series models a surprisingly accurate detection of bubbles in moderately dense bubble flow has been achieved. Although some of the presenation is, at least for me, a bit on a freak-level this depth of details may be helpful for the further developments in this dynamic branch. In general I recommend a publication of the manuscript and have only minor comments.

Line 94/95 are a repetion..

112-115 the same

141: improve readability: "..., 1 bubble confidence"

Fig. 7: To enable consistent reading, you should avoid abbreviations in the caption, i.e. mean square error instead of MSE

Is OK, some typos and redundancies should be removed (see comment)

Reviewer 4 Report

This paper presents the application of YOLO architectures to estimate bubble size. The current draft is mostly well written, the tool seems promising, and the topic is relevant in process engineering and measurements. The following comments must be taken into consideration before further reviewing:

1.- The authors do not include bubble viewers (typically used in froth flotation) to measure bubble size (e.g. McGill and Anglo Platinum BSA). Significant efforts have been made in flotation to avoid issues associated with depth of field, which also may reduce highly dense clusters. In addition, bubble sampling is also critical to reduce the presence of clusters. Please discuss and expand your literature review on these points.

2.- I am impressed that the following paper was not cited on the use of neural networks to characterize bubble size.

https://doi.org/10.1016/j.patrec.2017.11.014

3.- Although the depth of field was not discussed in this draft, a 2D representation for the bubbles was assumed (ellipse detection). I am not sure how the employed database considers a single 2D photograph, please refer to the design of the McGill BSA. In any case, the authors should take stereological corrections into consideration to determine bubble size. Please discussed and expand from literature.

4.- The title refers to bubble size distributions (BSDs); however, no BSDs are presented throughout the manuscript. In process engineering, more important than detecting and sizing individual bubbles, we are interested in accurately estimating bubble size distributions. Please compare the estimated versus actual BSDs, or present Kolmogorov-Smirnov tests.

5.- I think your performance indicators are not the best. In my view, it would be much better to plot actual dBs (bubble diameters) versus estimated dBs to determine the goodness of fit in the entire bubble size range. A high coef. of correlation does not deliver information on biases in specific size ranges.

6.- The authors claimed that "bubbles of larger size are easy to detect". I disagree, bubbles are easy to detect in spherical regimes and difficult to detect when large bubbles co-exist with small bubbles. Please refer to the following paper. Please discuss and expand.

https://doi.org/10.1016/j.mineng.2020.106636

7. The authors exaggerate when they say "the method proposed in this paper performs extremely well". Please be modest, as the tool was tested with ideal images.

8.- The images presented in Fig. 6 do not seem realistic. Apparently, they have been considered under ellipsoidal regimes (no circles, which implies operation with no frothers or any surfactants). In this regard, it is impossible to have such a low void fraction if the machine is not operated in spherical regimes (with high surfactant concentration), except for the case that some bubbles could be some layers behind the focused bubbles. In the last case, these second further bubbles should look smaller and out of focus. Please discuss or correct.

The English style is fine, but may be improved.

Round 2

Reviewer 1 Report

Author has made the changes as suggested and can be accepted in the present form 

Reviewer 2 Report

The manuscript has been improved after major revision. Therefore, I suggest it can be accepted after minor language editing.

Minor editing

Reviewer 4 Report

The authors should be congratulated for the quality of this piece of research.

Although I am not a native English speaker, I think the English is reasonably good and can be directly edited by MDPI personnel.